# Investigating Spatial Patterns of Tumor and Stroma in Gastric and Colorectal Cancer for Survival Prediction

Sanghoon Lee, Yellu Siri
*Department of Computer Science*
*Kennesaw State University*
Marietta, GA, USA

Sung Hak Lee
*Department of Hospital Pathology*
*Seoul St. Mary's Hospital*
Seoul, Korea

Jae-Ho Cheong
*Department of Surgery*
*Yonsei University College of Medicine*
Seoul, Korea

Minji Kim, Sunho Park, Tae Hyun Hwang
*Section Surgical Research*
*Vanderbilt University Medical Center*
Nashville, TN, USA

*Abstract*—The spatial patterns of tumor, stroma, and tumor-infiltrating lymphocytes (TILs) within the tumor microenvironment significantly affect cancer progression and are associated with clinical outcomes. Understanding the exact significance and statistical implications of these spatial patterns is, however, challenging due to the complexity of spatial interactions. In this paper, we investigate the spatial patterns related to patient survival in gastric and colorectal cancer using four classifiers to predict tumor, stroma, TILs, and Microsatellite Instability (MSI) status, along with the Getic-Ord-Gi* statistic as the spatial image analysis, analyzing four large patient cohorts. U-Net, a deep learning model for semantic image segmentation, was used to predict tumor, stroma, and TILs in digitized Hematoxylin and Eosin-stained formalin-fixed paraffin-embedded (FFPE) sections, and ResNet-18 was employed to predict the MSI status. The Getis-Ord-Gi* statistic was applied to determine statistically significant tumor hotspot regions by examining their relationship to surrounding areas. Kaplan-Meier analyses and log-rank tests were performed to assess the association between the spatial patterns of tumor and stroma and overall survival in the patient cohorts. Results indicate that the stroma composition around tumor hotspot regions, identified by the Getis-Ord-Gi* statistic, shows significant differences in overall survival among patients with gastric and colorectal cancer. The log-rank test was included to examine the relationship between stroma composition and MSI and ACTA2 expression levels.

*Index Terms*—stroma, tumor, tumor-infiltrating lymphocytes, microsatellite Instability, spatial image analysis, deep learning, and whole slide image.

## I. INTRODUCTION

Gastric and colon cancers are major global health burdens, ranking among the top causes of cancer-related death. [1]. While current treatments, including chemotherapies, immunotherapy, and targeted therapies, interact with various cellular components, their efficacy is significantly influenced by the tumor microenvironment (TME) [2]. A growing body of evidence highlights the importance of understanding the ecological processes within tumors, particularly the docu-mented spatial variations of cancer components [3]. This spatial heterogeneity of cell components in the TME has significant clinical implications [4]. Therefore, investigating the spatial patterns and their relationship to patient survival in gastric and colorectal cancers is crucial for optimizing treatment strategies.

Recent computational advances have revolutionized the analysis of large digitized histological sections, enabling the automated identification and quantification of cellular contexts within complex tumor microenvironments [5]. This has significantly advanced our understanding of immune cell spatial patterns and their clinical importance, even allowing for the correlation of immune infiltration with cancer prognosis. However, many studies have overlooked the crucial interactions of other cellular components, particularly stromal cells. While some research has tracked specific stromal cell types like Carcinoma-Associated Fibroblasts (CAFs) for their tumor-promoting roles, these efforts have not fully explored the spatial ratio of stromal cells around tumor hotspots in whole slide images or leveraged spatial statistics. Though limited, existing studies suggest that stromal changes and ACTA2 expression are linked to gastric cancer development and patient survival [6], [7]. Similarly, ACTA2 expression has shown therapeutic significance in stroma-rich colon cancer [8], [9]. This collective evidence highlights the need for further investigation into the relationship between ACTA2 gene expression and the spatial composition of stromal cells around tumor hotspots.

In this paper, we explore the spatial patterns in gastric and colorectal cancers across four patient cohorts: TCGA-STAD (N=347), TCGA-COAD (N=428), YONSEI (N=614), and STMARY (N=233). We created three U-Net deep learning classifiers to identify tumor, stroma, and tumor-infiltrating lymphocyte (TIL) regions, and a ResNet-18 classifier to determine microsatellite instability (MSI) status. These models were applied to 1,622 digitized Hematoxylin and Eosin-

stained formalin-fixed paraffin-embedded (FFPE) sections. Subsequently, we analyzed stromal cell composition within the tumor hotspot regions, identified using the Getis-Ord-Gi* statistic.

## II. RELATED WORKS

Spatial image analysis is a powerful technique that examines the relationships between neighboring pixels in image data. In medical imaging, it helps us understand and model the unique characteristics of different cell types within their microenvironment. For example, Wang et al. [10] showed how quantifying histomorphometric features through spatial relationships can reveal complex interactions in the tumor microenvironment. Similarly, Karimi et al. [11] demonstrated that spatial cellular neighborhoods can impact patient survival in glioblastoma. A common and accurate approach in spatial analysis involves creating a density map, which often relies on region prediction. Among these methods, the Getis-Ord-Gi* statistic is a widely used hotspot analysis technique, identifying significant hotspot regions by assessing the relatedness of a feature to its neighbors. Its robust mapping strategy for spatial phenomena has led to its adoption in numerous studies [12]–[14]. Here, we are using the Getis-Ord-Gi* statistic to uncover key characteristics of image features within the tumor microenvironment.

Our paper leverages two popular deep learning models, U-Net [15] and ResNet-18 [16], to power four classifiers for predicting tumor, stroma, TILs, and MSI status, respectively, across 1,622 digitized Hematoxylin and Eosin-stained whole slide images. We use U-Net, a semantic image segmentation model, because it excels at pixel-wise classification. Unlike traditional image classification, which assigns a single category to an entire image, U-Net precisely labels each pixel with its specific class. This capability has made U-Net and similar models invaluable across various fields, including medical image analysis, autonomous vehicles, and pedestrian detection [17]–[19].

## III. MATERIALS AND METHOD

Our study utilized 1,622 digitized whole slide images (WSIs), totaling approximately 1,000 gigabytes, from hematoxylin and eosin-stained (H&E) formalin-fixed paraffin-embedded (FFPE) sections across four patient cohorts. Two cohorts, TCGA-STAD (N=347) and TCGA-COAD (N=428), were openly accessed from the National Cancer Institute's Genomic Data Commons data portal. The remaining two patient cohorts, YONSEI (N=614) and STMARY (N=233), were collected from Yonsei University College of Medicine and Seoul St. Mary's Hospital, respectively, both located in Seoul, Republic of Korea. All these data were used for tissue image prediction, spatial analysis, and subsequent statistical analysis. These whole slide images were initially split into smaller, manageable tiles (4096x4096 pixels at 0.5 microns per pixel (MPP)). To minimize color variability and ensure consistency across the diverse datasets, all images underwent

color normalization using the Reinhard method. The details are shown in Fig. 1A.

Diagnostic H&E-stained slides, obtained in SVS format, underwent meticulous preparation. We created four deep learning classifiers (tumor, stroma, TILs, and MSI status) using the U-Net semantic segmentation model and ResNet-18 deep learning model. The U-Net was trained on the Crowdsourcing dataset (151 images) [20], leveraging its well-documented performance in similar research [21], [22]. To ensure consistency and reduce color variability, each slide was tessellated into 256x256 pixel tiles at 20x magnification (0.5 MPP) and normalized using Reinhard normalization based on performance evidence [23]. Model training involved 50 epochs, the Adam optimizer with a 0.001 learning rate, a batch size of 16, a binary cross entropy loss function, and L1 regularization. Additionally, an MSI classifier, previously trained on ResNet18, was employed [24]. The details are shown in Fig. 1B.

To pinpoint tumor hotspot regions within the predicted whole slide images, we applied Getis-Ord-Gi* statistic. This statistical method is particularly effective for identifying statistically significant hotspot features by evaluating their relatedness to its neighboring values. Specifically, it measures the degree of spatial clustering of a feature (in our case, tumor density) by comparing the local sum of the feature values to the global sum. We employed a K-nearest neighbor approach with a 5x5 grid and 10 neighbors as the spatial weights matrix. The choice of neighbor count significantly influences the analysis, as it determines which features contribute to the local sum for each feature. The Getis-Ord Gi* statistic is formally presented as:

$$G_i^* = \frac{\sum_{j=1}^{n} w_{i,j} x_j - \overline{X} \sum_{j=1}^{n} w_{i,j}}{S \times \sqrt{\frac{n \sum_{j=1}^{n} w_{i,j}^2 - (\sum_{j=1}^{n} w_{i,j})^2}{n-1}}} \quad (1)$$

where $w_{i,j}$ is the weight element between features $i$ and $j$. $x_j$ represents the value of feature $j$. $\overline{X} = \frac{\sum_{j=1}^{n} x_j}{n}$, the average of the feature values. $n$ is the total number of features. $S = \sqrt{\frac{\sum_{j=1}^{n} x_j^2}{n} - (\overline{X})^2}$, the standard deviation of the feature values.

We identified tumor hotspot regions in the predicted WSIs by calculating z-scores from the Getis-Ord-Gi* statistics. A high positive z-score indicates a hot spot, signifying dense tumor regions, while a negative z-score indicates a cold spot. We then correlated these identified tumor hot spots with stroma composition and clinical outcomes using Kaplan-Meier analysis and log-rank tests to assess statistical significance within the patient cohorts. All statistical tests with a p-value less than 0.05 were considered statistically significant. All statistical analyses were performed using the R package. The details are shown in Fig. 1C.

## IV. EXPERIMENT RESULTS

In this section, we describe our findings in the investigation of stoma composition around tumor hotspot regions in four

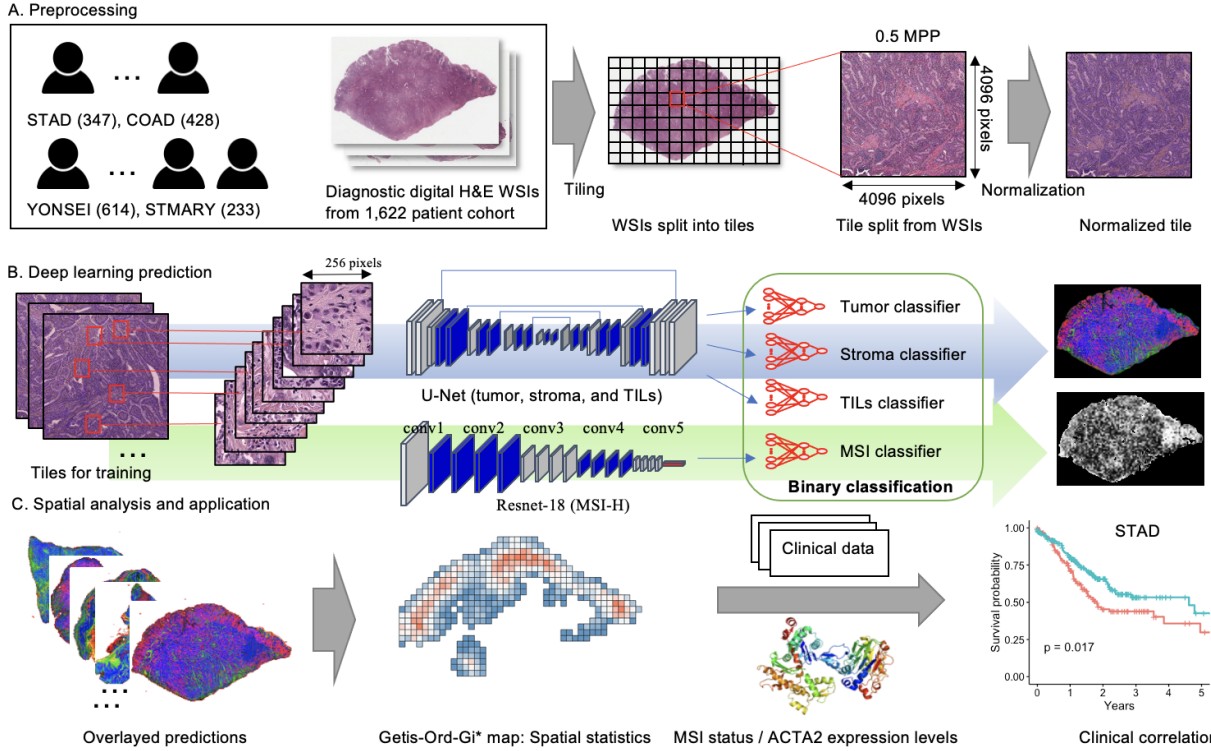

Fig. 1.  Overview of the proposed method. A. Whole slide image preprocessing. B. Tissue class prediction. C. Spatial analysis and clinical correlation.

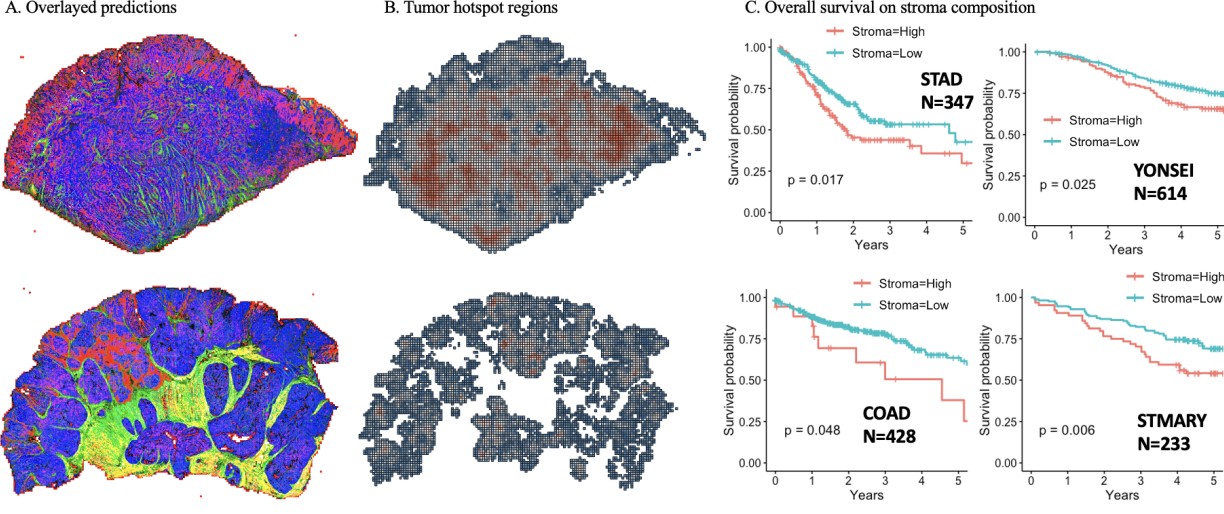

Fig. 2.  A. Overlayed regions for tumor (blue), stroma (green), and TILs (red) predicted by U-Net classifiers Gastric cancer (top) and colon cancer (bottom). B. Tumor hot region generated by Getis-Ord-Gi* (Z-score mean: 2.8016). The region with the red represents the tumor hot region (tumor abundance) Gastric cancer (top) and colon cancer (bottom). C. Overall survival on four patient cohorts: STAD (top left), YONSEI (top right), COAD (bottom left), and STMARY (bottom right).

patient cohort. First of all, the stroma composition surrounding the tumor hot region, spatially generated by the Getis-Ord-Gi* statistic, showed a significant difference in overall survival on four patient cohort (TCGA-STAD, TCGA-COAD, YONSEI, and STMARY). Compared to the high stroma group, the low stroma group showed a better survival probability (p-value=0.017 in TCGA-STAD, p-value =0.048 in TCGA-COAD, p-value=0.025 in YONSEI, and p-value=0.006 in STMARY) in overall survival. The Kaplan-Meier plots for four patient cohort are shown in Fig. 2C and clinical characteristic of patients in the TCGA-STAD and TCGA-COAD is described in Table 1. Examples of overlayed predicted results of whole slide images and their tumor hot regions generated by the Getis-Ord-Gi* statistic across gastric cancer and colon cancer are shown in Fig. 2 and the examples of tumor hotspot regions based on the z-scores are shown in Fig. 6.

TABLE I
CLINICAL CHARACTERISTICS OF PATIENTS IN THE TCGA-STAD AND TCGA-COAD

| Metadata | TCGA-STAD (n=347, high/low: 128/219) | |
| --- | --- | --- |
| | Stroma high | Stroma low |
| Average age | 64.95 | 65.18 |
| Female | 41 (32.0%) | 79 (36.1%) |
| Male | 87 (68.0%) | 140 (63.9%) |
| | TCGA-COAD (n=428, high/low: 18/410) | |
| Average age | 65.83 | 66.99 |
| Female | 9 (50.0%) | 200 (48.8%) |
| Male | 9 (50.0%) | 210 (51.2%) |

Second, our investigation explored the relationship between stromal composition within tumor hotspots, identified using the Getis-Ord-Gi* statistic, and Microsatellite Instability (MSI) status. We quantified the percentage of high and low stromal composition across four patient cohorts: TCGA-STAD (N=346), TCGA-COAD (N=410), YONSEI (N=575), and STMARY (N=233), stratified by MSI-High (MSI-H) and Microsatellite Stable (MSS) statuses. Our findings reveal that high stroma composition is consistently low in MSI-H tumors across all four cohorts (10%, 0%, 5%, and 15% for TCGA-STAD, TCGA-COAD, YONSEI, and STMARY, respectively). Conversely, high stromal composition is markedly prevalent in MSS tumors (90%, 100%, 95%, and 85% for TCGA-STAD, TCGA-COAD, YONSEI, and STMARY, respectively). To assess the clinical significance of these observations, we conducted Kaplan-Meier survival analysis with log-rank tests, comparing four distinct groups based on the combination of stroma composition (high/low) and MSI status (MSI-H/MSS). The survival curves demonstrated that incorporating stromal composition, derived from spatial image analysis, yielded a more significant differentiation in patient survival compared to relying solely on MSI status. Further details are presented in Fig. 3.

Thirdly, we investigated the potential link between stromal composition within Getis-Ord-Gi* generated tumor hotspots and ACTA2 gene expression. Given a recent study [7] highlighting ACTA2's independent association with gastric cancer

TABLE II
CLINICAL CHARACTERISTICS OF PATIENTS IN THE TCGA-STAD AND TCGA-COAD WITH ACTA2 EXPRESSION.

| | STAD (n=288) | | | |
| --- | --- | --- | --- | --- |
| | SH-AH(83) | SL-AH(127) | SH-AL(29) | SL-AL(49) |
| Average age | 65.6 | 65.39 | 63.62 | 65.55 |
| Female | 28 (34%) | 47 (37%) | 8 (28%) | 19 (39%) |
| Male | 55 (66%) | 80 (63%) | 21 (72%) | 30 (61%) |
| | COAD (N=403) | | | |
| | SH-AH(8) | SL-AH(127) | SH-AL(9) | SL-AL(259) |
| Average age | 67.75 | 65.18 | 66.33 | 67.32 |
| Female | 5 (62.5%) | 62 (49%) | 3 (33%) | 122 (47%) |
| Male | 3 (37.5%) | 65 (51%) | 6 (67%) | 137 (53%) |

patients overall survival, we aimed to explore this further interaction. To achieve this, we conducted log-rank test on Kaplan-Meier survival curves to compare the time-to-event distributions across four distinct groups: high stroma/high ACTA2 (SH-AH), high stroma/low ACTA2 (SH-AL), low stroma/high ACTA2 (SL-AH), and low stroma/low ACTA2 (SL-AL). This analysis was performed on three patient cohorts, TCGA-STAD (N=288), TCGA-COAD (N=403), and YONSEI (N=63), with available bulk sequencing gene expression data. Our results demonstrate a statistically significant difference in group survival when incorporating spatial stromal composition alongside ACTA2 (p-value: 0.0089 for TCGA-STAD, p-value: 0.0047 for TCGA-COAD, and p-value: 0.05 for YONSEI). This indicates that spatial stromal analysis provides a more powerful prognostic factor than ACTA2 expression alone. Furthermore, a focused log-rank test comparing the SH-AH and SL-AL groups revealed highly significant differences in overall survival (p-value: 0.0013 for TCGA-STAD and p-value: 0.0021 for TCGA-COAD, and p-value: 0.0022 for YONSEI). The detailed findings are further illustrated in Fig. 4 and Table 2.

## V. DISCUSSION

Previous spatial analysis studies in gastric tumor microenvironments primarily focused on the distribution of specific gene expression or the interactions between cancer and immune cells, often using non-globally applicable distribution techniques [25], [26]. In contrast, our study delves into the stromal cell composition within tumor hot regions, leveraging the Getis-Ord-Gi* statistic to identify statistically significant areas. This approach is critical given evidence suggesting that increased stromal involvement can worsen patient prognosis [27]. Our findings reveal two major associations: stromal composition is significantly linked to MSI status and stromal composition also correlated with ACTA2 gene expression levels. By analyzing stroma distribution around tumor hotspots in MSI-High (MSI-H) and Microsatellite Stable (MSS) group across four large patient cohorts (TCGA-STAD, TCGA-COAD, YONSEI, and STMARY), we observed a consistently low percentage of high stroma in MSI-H tumors, contrasted by a high percentage in MSS tumors. Furthermore, Kaplan-Meier survival analysis demonstrated that incorporating stro-

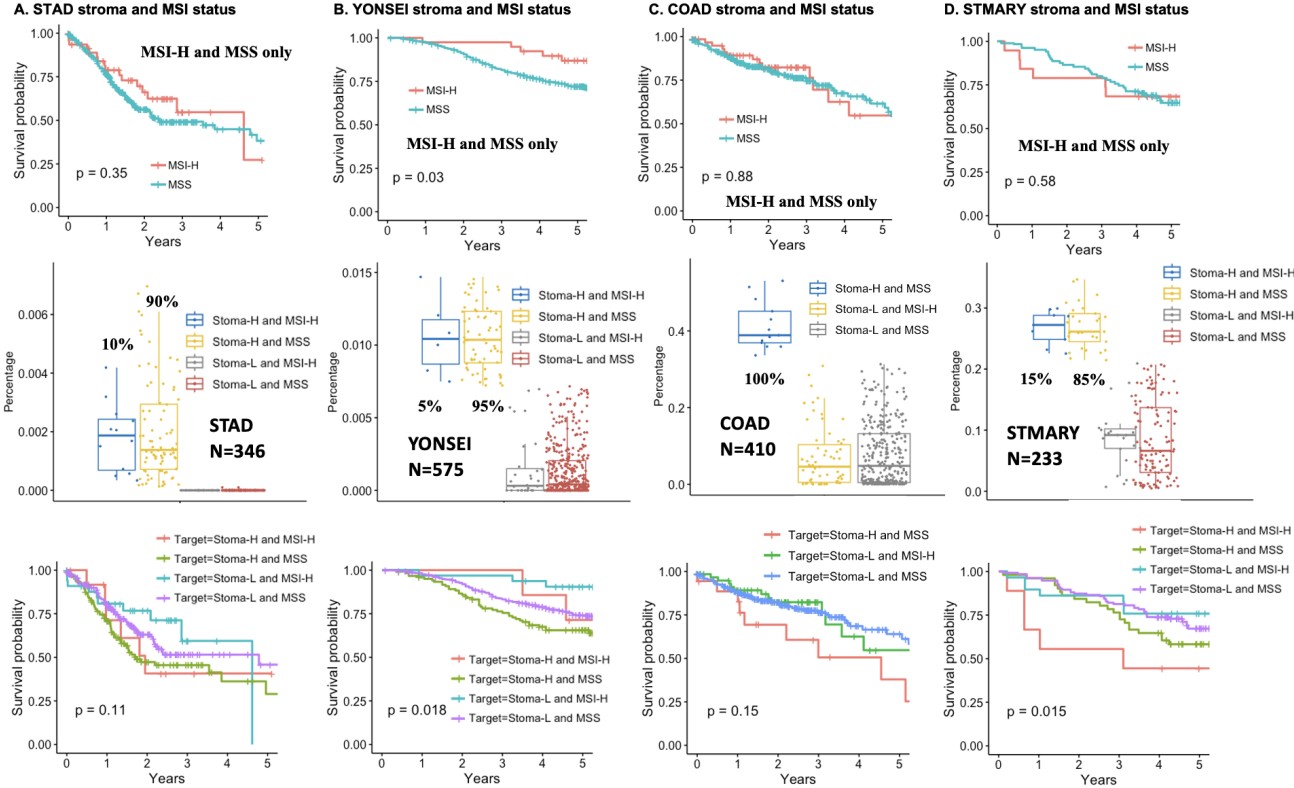

Fig. 3. Spatial statistics linked to the patient survival. A. TCGA-STAD stroma composition with MSI status. B. YONSEI stroma composition with MSI status. C. TCGA-COAD stroma composition with MSI status. D. STMARY stroma composition with MSI status. Kaplan-Meier survival curves with MSI status, the percentage of Stroma high/low and MSI-H/MSS, and Kaplan-Meier survival curves with both strmoal compositions and MSI status (top to bottom).

mal composition yielded a more significant differentiation in patient survival than MSI status alone. Our experiments on TCGA-STAD, TCGA-COAD, and YONSEI cohorts revealed statistically significant survival differences among four groups defined by high/low stroma composition and high/low ACTA2 expression. Notably, a significant survival disparity was observed between groups with high stroma/high ACTA2 and low stroma/low ACTA2 [7]. Overall, our study demonstrates that the spatial distribution of stromal cells within the tumor microenvironment, as predicted from H&E-stained whole slide images using deep learning models and analyzed with the Getis-Ord-Gi* statistic, is a crucial prognostic factor for patient survival.

## VI. CONCLUSION

In this study, we investigated stromal cell composition within tumor hotspot regions in gastric and colorectal cancer whole-slide images. Utilizing deep learning classifiers, we accurately predicted tumor, stroma, TILs, and MSI status regions across large patient cohorts (TCGA-STAD, TCGA-COAD, YONSEI, and STMARY). The Getis-Ord-Gi* statistic was then employed to identify tumor hotspots, around which stromal cell composition was assessed. Our findings robustly demonstrate that stromal cell composition around tumor hotspots is significantly associated with patient survival prognosis in gastric and colorectal cancers. Furthermore, we

also showed that integrating stromal composition with ACTA2 gene expression levels yields a more significant difference in patient survival, suggesting a valuable prognostic marker. These results hold promising potential for improving patient prognosis identification in gastric and colorectal cancers.

However, our current work has certain limitations that warrant future investigation. Specifically, we did not compare our findings against recent deep learning-based survival models (e.g., DeepSurv and attention-based MIL models) [28], [29] using the same datasets. This comparison would strengthen the evaluation of our spatial analysis method. Additionally, while our survival analyses show promising trends, we acknowledge the absence of multiple-hypothesis correction, which carries a potential risk of over-interpreting marginal results, particularly within the YONSEI dataset. Lastly, our analysis currently does not account for additional patient-level variables such as disease stage, therapy received, and detailed demographics. Addressing these limitations will be a focus of our future work.

## ACKNOWLEDGMENT

This research was supported by the National Science Foundation under Grant No. 2409705.

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
