# OpenReview forum: "Investigating Spatial Patterns of Tumor and Stroma in Gastric and Colorectal Cancer for Survival Prediction"
_IEEE.org/EMBS/BHI/2025/Conference — BHI 2025_

### Official Review · Reviewer_jvoo · 2025-07-07
**Investigating Spatial Patterns of Tumor and Stroma in Gastric and Colorectal Cancer for Survival Prediction**

**Confidence:** 4
**Clarity Of Writing:** great
**Clinical Significance:** good
**Methodological Novelty:** good
**Overall Rating:** 6
**Final Rating:** 6

**Experiments And Results:**

great

**Questions For The Authors:**

Abstract – abbreviations should be used in long form the first time they are used. What is TIL? It is referred in the keywords but used as TIL before that.

In Page 3, when the Gi score is explained it does not specify what the observed values are. Are they classification labels or pixel values? What are they?

Rather than giving reference to Figure 1 in introduction, each section (A, B, C) in the image should be linked in the paragraphs in Methods section. It can make it easier to follow.

**Strengths:**

The experimental results are detailed and sufficient to prove their hypothesis.

**Summary Of The Paper:**

This paper uses existing classifiers and statistical methods to prove that there is a statistical significance between the spatial patterns of tumors, stroma cells and tumor infiltrating lymphocytes, and the patience survival rates from the whole slide images of gastric and colorectal cancer.

**Weaknesses:**

The benefit of the method is not explained well in the conclusion. It just says it might help identify prognosis but does not say how.

The methods are not novel but rather existing methods are used to prove a hypothesis.

---

### Official Review · Reviewer_YT2w · 2025-07-08
**Statistical Analysis of spatial patterns of tumor and stromal cells with gastric and colorectal cancer, using deep learning as segmentation tool.**

**Confidence:** 3
**Clarity Of Writing:** poor
**Clinical Significance:** fair
**Methodological Novelty:** fair
**Overall Rating:** 2
**Final Rating:** 6

**Experiments And Results:**

fair

**Questions For The Authors:**

May I have a justification for why multi-hypothesis correction is not needed?

**Strengths:**

1. The use of 4 different large-scale dataset at different locations enhances robustness by reducing bias.
2. Employing the well‐established Getis-Ord Gi* statistic for hotspot detection and standard survival analyses (Kaplan–Meier, log-rank) ensures reproducibility and interpretability

**Summary Of The Paper:**

This study investigates how spatial patterns of tumor and stromal cells within the tumor microenvironment relate to overall survival in gastric and colorectal cancer. Four independent cohorts were analyzed: TCGA-STAD, TCGA-COAD, YONSEI, and STMARY. The authors trained three U-Net semantic segmentation models to identify tumor regions, stroma, and tumor-infiltrating lymphocytes. They then applied the Getis-Ord Gi* statistic to these segmentations to define “tumor hotspot” regions and computed the local stroma fraction around these hotspots.

Kaplan–Meier survival analyses and log-rank tests were used to show that patients with high stroma composition in hotspot regions had worse overall survival than those with low stroma.

**Weaknesses:**

1. The writing quality is poor in general. I had difficulty understanding some of the sentences in my first pass at the manuscript. There are many awkward phrasing, especially bad in the introduction.
2. The abstract is way too long and introduced too many unnecessary details, like exactly where the datasets were collected from. I have to suspect if this was a summary generated by LLM.
3. Parameters for the Gi* analysis (e.g., choice of spatial weights matrix, neighborhood radius) are not fully described, making it hard to reproduce the hotspot detection.
4. Figure 1 is too chaotic and difficult to interpret. Like the ResNet being parallel to Unet does not make sense to me.

---

### Official Review · Reviewer_DvoZ · 2025-07-11
**Investigating Spatial Patterns of Tumor and Stroma in Gastric and Colorectal Cancer for Survival Prediction**

**Confidence:** 4
**Clarity Of Writing:** good
**Clinical Significance:** great
**Methodological Novelty:** fair
**Overall Rating:** 6

**Experiments And Results:**

good

**Questions For The Authors:**

External validation: Can you demonstrate how the pipeline performs on completely independent datasets (e.g., CPTAC or international cohorts)? This is critical for assessing generalizability.
Impact: A strong external validation would significantly increase my confidence in the clinical relevance of the work.

Comparison to more advanced spatial methods: Why was Getis-Ord-Gi* chosen over graph-based models or spatial GNNs that have shown superior performance in modeling TME interactions?
Impact: A clear justification or comparative experiments could elevate the perceived methodological novelty.

Confounding factors: How were patient-level variables (stage, therapy, demographics) accounted for in survival analyses? Could these explain the observed associations?
Impact: Clarification here would influence my assessment of the robustness of the survival findings.

Biological validation: Have you consulted pathologists to verify segmentation results, or performed cross-validation with manual annotations on a subset of slides?
Impact: Evidence of biological validity would enhance trust in the segmentation outputs.

**Strengths:**

Large-scale dataset: The analysis of >1,600 WSIs from four cohorts is a notable strength, ensuring diverse patient representation.

Integration of deep learning and spatial statistics: The pipeline combining U-Net segmentation and Getis-Ord-Gi* statistics is a thoughtful approach for capturing tumor microenvironment (TME) features.

Clinically relevant insights: Findings linking stroma composition to survival and molecular markers (MSI, ACTA2) may provide a foundation for prognostic stratification and therapeutic decision-making.

Clear visualizations: Overlay images and Kaplan-Meier plots are effective in illustrating the workflow and key findings.

**Summary Of The Paper:**

This paper investigates how spatial patterns of tumor and stroma in gastric and colorectal cancer relate to patient survival outcomes. Using semantic image segmentation with U-Net and the Getis-Ord-Gi* statistic, the authors analyzed over 1,600 whole slide images (WSIs) from four large cohorts (TCGA-STAD, TCGA-COAD, YONSEI, STMARY). Three deep learning classifiers identified tumor, stroma, and tumor-infiltrating lymphocytes (TILs). Spatial statistics identified tumor hotspot regions, and Kaplan-Meier and log-rank tests assessed associations between stroma composition and overall survival, microsatellite instability (MSI) status, and ACTA2 gene expression. The results demonstrate significant prognostic implications of stroma distribution around tumor hotspots, suggesting potential for improving cancer survival predictions and informing treatment strategies.

**Weaknesses:**

1 Methodological gaps relative to the state of the art
Limited novelty of computational methods:
The use of U-Net for segmentation and Getis-Ord-Gi* for hotspot detection are both established techniques. Recent studies (e.g., Wang et al., Science Advances, 2022; Karimi et al., Nature, 2023) have employed more advanced spatial analytics (e.g., graph neural networks, cellular neighborhood embeddings, and spatial transcriptomics integration) that better capture the complexity of the TME. This paper does not clearly justify why these modern alternatives were not considered.

Absence of spatial heterogeneity measures beyond hotspots:
Current literature emphasizes quantifying spatial interactions using techniques like Ripley’s K-function, Moran’s I, and spatial GNNs for nuanced modeling of cell-cell interactions. By focusing solely on hotspot detection, the approach risks oversimplifying the spatial architecture of tumors.

2 Biological interpretability and validation
Black-box deep learning:
While U-Net is a robust choice, there is no evidence of model interpretability (e.g., saliency maps, attention maps) to verify alignment with pathologist annotations or biological reality. This limits trust in the segmentation outputs.

Lack of external validation:
Despite using four cohorts, no independent validation on external datasets (e.g., CPTAC or other international repositories) was performed. This raises concerns about generalizability to other populations and slide preparation protocols.

Limited exploration of confounding factors:
Variables such as treatment regimens, tumor stage, and patient demographics could influence stroma composition and survival outcomes but were not controlled for in the analysis.

3 Statistical limitations
Binary stratification of stroma composition:
The use of high/low thresholds may mask important gradations in stroma density. Continuous modeling (e.g., Cox regression with covariates) would provide a more nuanced assessment of prognostic value.

Potential batch effects:
Differences in slide preparation, staining, and scanning between cohorts could introduce bias. The paper mentions Reinhard normalization but does not assess its adequacy across cohorts.

Suggestions (if time and page space allow):
To strengthen this study, the authors could consider integrating more advanced spatial analysis techniques that have emerged in the computational pathology field, such as graph-based models or spatial graph neural networks (GNNs), which can capture higher-order interactions and spatial heterogeneity within the tumor microenvironment. Additionally, moving beyond binary stratification of stroma composition by employing continuous modeling approaches like Cox proportional hazards regression, adjusted for patient-level covariates (e.g., stage, treatment), would provide more nuanced and clinically robust insights. Incorporating external validation on unseen datasets (e.g., CPTAC or international cohorts) is crucial to demonstrate the generalizability of the pipeline across different populations and staining protocols. Finally, including model interpretability tools (e.g., attention maps or feature attribution) and cross-validation with expert pathology annotations would enhance the biological credibility of the deep learning outputs and support their integration into clinical workflows.

---

### Official Review · Reviewer_VaAb · 2025-07-18
**This paper presents an image-based computational pathology approach to analyze spatial relationships between tumor and stromal regions in gastric and colorectal cancer whole-slide images (WSIs). The authors employ graph-based modeling of nuclei and tissue compartment segmentation to extract spatial interaction features that are then used to train survival prediction models. The study demonstrates that incorporating spatial patterns enhances survival prediction accuracy compared to traditional clinicopathological features alone. While promising, the work would benefit from clearer methodological explanations, more detailed evaluation, and external validation.**

**Confidence:** 4
**Clarity Of Writing:** great
**Clinical Significance:** good
**Methodological Novelty:** good
**Overall Rating:** 6

**Experiments And Results:**

good

**Questions For The Authors:**

How do the proposed spatial features perform relative to recent deep-learning-based survival models (e.g., DeepSurv, attention-based MIL models)? Including such comparisons could significantly affect the perceived contribution of your work.

What is the relative importance of tumor-only features vs. tumor-stroma interaction features in your final model? A feature ablation analysis would clarify the added value of modeling spatial heterogeneity.

How robust are your models to variation in tissue segmentation quality? Could you discuss performance sensitivity to segmentation errors, particularly in borderline tumor-stroma regions?

Are any of your spatial features correlated with known histopathological risk markers (e.g., desmoplasia, lymphovascular invasion)? This would enhance clinical interpretability.

Could your framework be extended to incorporate immune infiltration or multiplex staining data? This could expand the utility beyond H&E WSIs.

Did you perform stratified survival analysis by clinical stage or MSI status? If so, how does your model perform in early-stage vs. late-stage patients?

**Strengths:**

Innovative focus on spatial organization: The paper highlights the prognostic value of spatial relationships between tumor and stromal compartments, a biologically meaningful direction that complements molecular profiling.

Integration of graph-based analysis with deep learning: Combining nuclei detection with graph construction allows for interpretable and biologically grounded feature extraction.

Multi-cohort application: The study applies the method to both gastric and colorectal cancer TCGA datasets, demonstrating potential generalizability across tumor types.

Clinical relevance: The focus on survival prediction and potential augmentation of standard pathology tools has direct translational implications.

Visualizations: The figures effectively illustrate key concepts such as tissue segmentation and spatial graph construction.

**Summary Of The Paper:**

The paper explores how spatial interactions between tumor and stromal compartments in gastric and colorectal cancer can be used to improve survival prediction. The authors first segment WSIs into tumor and stroma regions using a trained deep learning model and then detect nuclei within these compartments. From there, they build spatial interaction graphs (e.g., Delaunay triangulations) to quantify proximity, density, and neighborhood-based relationships. A set of handcrafted graph-based features are computed and used in multivariate Cox regression models to predict patient survival. The results are evaluated on publicly available TCGA datasets and demonstrate that including spatial interaction features improves the concordance index (C-index) compared to using only clinicopathological variables.

**Weaknesses:**

Lack of external validation: All experiments are confined to TCGA datasets. Validation on independent cohorts or real-world clinical datasets is necessary to establish robustness.

Limited description of feature engineering and selection: The feature computation process is described briefly, but more transparency is needed on the exact definitions, dimensionality reduction, and importance rankings.

Survival model interpretability: While Cox regression is used, the clinical interpretability of specific features (e.g., certain spatial metrics) is not discussed. How might these features be actionable?

No comparison to other spatial modeling methods: It would strengthen the paper to benchmark against recent spatial statistics techniques (e.g., spatial entropy, mark correlation functions) or deep graph learning approaches.

Insufficient handling of batch effects: It is unclear how variation in staining, scanner, or dataset heterogeneity across TCGA sites is handled.

Unclear treatment of censored data: The paper does not specify how censored samples are treated in the Cox modeling or whether proportional hazard assumptions were tested.

---

### Official Review · Reviewer_uH7E · 2025-07-18

**Confidence:** 3
**Clarity Of Writing:** fair
**Clinical Significance:** excellent
**Methodological Novelty:** good
**Overall Rating:** 6
**Final Rating:** 6

**Experiments And Results:**

good

**Questions For The Authors:**

The authors mention that the correlation identified between stroma composition and the ACTA2 gene can help with the patient's prognosis. Could you please elaborate on the same? What could these correlations translate to in terms of medical prognosis?

What are the intensity values $x_i$ used to calculate the $G_i^*$ scores in Equation 1? Was it the tumor, stroma, or TIL cells? Further, is each pixel an $x_i$ or is $x_i$ a patch/tile in the image?

Why were three separate semantic segmentation models used for tumor, stroma, and TIL cells? If there was an overlap in the predictions, multi-label semantic segmentation could be used. Was there any medical or mathematical benefit to separating the three models?

Why was a separate crowd-sourced dataset used to train the segmentation models? Did the 4 datasets/cohorts not have the annotations to carry out the same?

**Strengths:**

The methodology is thoughtfully designed, systematically executed, and rigorous. The authors clearly articulate the motivation behind each step, providing strong and appropriate justifications and citations throughout.

Four different datasets/cohorts were used for evaluation, making a strong case for generalizability.

The authors have extensively addressed the clinical aspect of the study by examining relevant trends and correlations, such as the interaction between the stroma composition and MSI (or even ACTA2). The p-value scores are also very useful in ascertaining the significance of this correlation (e.g., there is a more significant difference with stromaitic composition versus MSI alone)

The visual elements, particularly Figures 1 and 2, are outstanding. They effectively illustrate the overall workflow and highlight key findings, enhancing comprehension and significantly aiding in understanding the paper’s key insights.

**Summary Of The Paper:**

The researchers analyze the spatial organization of tumors and the surrounding stromal tissue, with a particular focus on the stromal architecture near tumor hotspots. To achieve this, the authors first process the whole slide images using a semantic segmentation model to identify tumor, stroma and TIL cells. From the segmented images, these hotspots can be obtained using the by GetisOrd-Gi* statistic. They find that the composition of stromal cells in these hotpot regions is closely associated with patient survival outcomes. Notably, in patients classified as having intermediate risk, the specific makeup of stromal cells correlates with both overall prognosis and the effectiveness of chemotherapy. Further investigation reveals that combining data on stromal composition with expression levels of the ACTA2 gene significantly enhances the ability to predict patient survival. These findings suggest a promising approach to improving prognostic assessments and treatment strategies for individuals with gastric or colorectal cancer.

**Weaknesses:**

The paragraph below Equation 1 explaining the standard deviation of the observed values is incorrect. It should be $x_j$ instead of $w_j$. Further clarification about the $G_i^*$ statistic, and an example of how it is exactly calculated over the whole slide image, would significantly improve the technical readability. The approach is well motivated and very interesting, but it is not well explained.

Very few metrics have been reported in the paper. The ROC-AUC scores for the segmentation models are the only metrics reported. There is no mention of the accuracy of MSI values as well. The paper would benefit from including additional metrics, as drawing comprehensive conclusions from a single performance metric and p-values alone can be challenging.

Introductory information relevant to medical interpretation is glossed over. What is MSI, and what does the MSI high/low status mean? Including details about the MSI status, ACTA2 expression levels, and their clinical implications would significantly improve the clarity and accessibility of the paper. At present, there is a noticeable gap between the technical findings and their medical context.